# How can BERT Understand High-level Semantics?

**Meriem Beloucif**, **Chris Biemann**

Language Technology Group, Dept. of Informatics, MIN Faculty, Universität Hamburg
{beloucif, biemann}@informatik.uni-hamburg.de

## Abstract

Pre-trained language models (PTLMs), such as BERT, ELMO and XLNET, have yielded state-of-the-art performance on many natural language processing tasks. In this paper, we argue that, despite their popularity and their contextual nature, PTLMs are unable to correctly capture high-level semantics such as linking age to the date of birth and rich to net worth. These kind of semantically-based inferences are performed systematically by humans, which is why we would we assume that PTLMs with all their language capabilities are able to make similar predictions. We show in this position paper that PTLMs are really good at making predictions for taxonomic relationships, but fail at attribute-value semantics like in rich and net worth.

## 1 Introduction

Knowledge bases such as Wikidata [Vrandečić and Krötzsch, 2014] constitute a good source for solving multiple NLP application as they contain structured information [Annervaz *et al.*, 2018; Nakashole and Mitchell, 2015; Rahman and Ng, 2011; Ratinov and Roth, 2009]. For instance, if we take the task of question answering, using a knowledge base could help answering factoid questions straightforwardly, and would only require access to a single fact from the knowledge base. For example, if we would like to know "How old is Joe Biden?", one can simply access his Wikidata page, and extract his "date of birth". Similarly, to check "How rich is Jeff Bezos?", we only need to extract his *net worth* from Wikidata. However, a simple task like this one requires us to have tools that can match *net worth* with *rich* as having similar meaning. Despite this task being straight forward to perform manually, it is complicated to achieve automatically, due to various linguistic challenges. In our example, rich and net worth are not synonyms, and do not share the same part-of-speech tag since the former is an adjective and the latter in a noun. However, they tend to appear together in text: net worth occurs 34 times in the Wikipedia page of Jeff Bezos, while rich/-er/-est appears 35 times. One way to automatically induce this correlation would be to used the current state of the art for contextual word embeddings, such as BERT [Devlin *et al.*, 2019]. Pre-trained language models

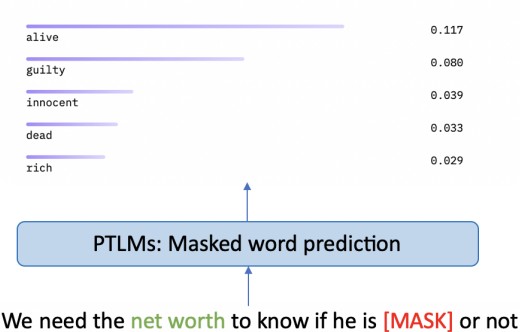

Figure 1: We purposefully MASK *rich* to check what would BERT predict given *net worth*. When looking for any rich celebrity, both Wikipedia and Wikidata would mention their net worth. Surprisingly, BERT is not able to associate *rich* with *net worth*, even after giving enough context for an accurate prediction. For our predictions and experiments, we used HuggingFace [Wolf *et al.*, 2020]
.

(PTLMs), such as BERT, have yielded state-of-the-art performance on many natural language processing tasks [Peters *et al.*, 2018a; Devlin *et al.*, 2019; Radford and Narasimhan, 2018], given their ability to create useful text representations. However, there has recently been a rising amount of research that uses probes to understand the level of syntax [Hewitt and Manning, 2019], semantics [Peters *et al.*, 2018b], linguistics [Belinkov *et al.*, 2017; Clark *et al.*, 2020; Tenney *et al.*, 2019] and commonsense [Petroni *et al.*, 2019; Davison *et al.*, 2019] that PTLMs encode.

In this position paper, we argue that, despite their popularity, PTLMs lack to capture the similarities between attributes and their qualitative values, which let people infer the attribute. We show simple cases by probing BERT [Devlin *et al.*, 2019]– using a masked-word-prediction based probing, where we mask an important word in a sentence such as a noun or an adjective, and give enough context for semantically similar words. BERT would predict what it believes most probably fills the mask, and thus exposing the capabilities of PTLMs to high-level semantics. For example, in Figure 1, we mask *rich* and give enough context by mentioning *net worth*. We note from the example that BERT fails at solving this simple problem since even when given enough

Table 1: The examples for probing BERT show that 6 out 19 times, BERT could predict the right information. These cases are mostly hypernyms and hyponyms. However, BERT fails to predict other semantic relations.

| Masked Sentence | Prediction | Truth |
|---|---|---|
| What's heavier or [MASK], gold or silver ? | **lighter**, heavier, stronger, thinner, weaker | denser |
| What's heavier and [MASK], gold or silver ? | stronger, lighter, heavier, darker, longer | denser |
| What's heavier, and thus has a higher [MASK], gold or silver? | value, grade, weight, price, quality | density |
| To know which is heavier, gold or silver, you need their [MASK] | :,to', . .,' , ; | density |
| We need the [MASK], to determine which is heavier, gold or silver | balance, test, gold, same, ratio | density |
| [MASK] such as gold, silver and copper | **metals**, elements, metal, commodities, materials | **metals** |
| Gold, [MASK] and iron | **silver**, copper, tin, nickel, platinum | **silver** |
| Which is bigger, and thus has a larger [MASK], Brazil or Argentina? | population, territory, **area**, market, influence | **area** |
| Which is larger, and thus has a higher [MASK], Brazil or Argentina? | population, influence, density, **territory**, advantage | **area** |
| We need the [MASK], to determine which is larger, Brazil or Argentina . | map, data, balance, ratio, maps | area |
| Brazil, [MASK] and Argentina | **Uruguay**, Paraguay, Chile, Bolivia, Colombia | **Uruguay** |
| Which is larger or [MASK], Brazil or Argentina? | **smaller**, lesser, less, shorter, small | **smaller** |
| What's bigger, and thus has a higher [MASK], London or Paris? | profile, price, prestige, ranking, value | population |
| Which city is denser or [MASK], Hong Kong or New York ? | hotter, faster, cooler, smaller, shorter | larger |
| Which city is [MASK], thus has more population per inhabitant ? | ''''', larger, bigger, smaller, greater | denser |
| Rabat is denser, thus it has more [MASK] | '',', air, water, land, density | inhabitant |
| We need [MASK] to determine which is denser, Hong Kong or New York . | then, to, , , data, again | population |
| We need [MASK] to determine which is denser, Delhi or New York . | time, only, data, information, now | population |
| We need [MASK] to determine which is denser, Delhi or Mumbai . | time, data, information, only, now | population |

context the closest prediction is *close*, and *rich* is not part of the five most likely predictions.

## 2 Semantically probing PTLMs

Large pre-trained language models have been shown to have outstanding performance in multiple downstream tasks in NLP. With fine-tuning, which is simply the retraining of a pre-trained model with fewer task-oriented examples, PTLMs gained even more popularity. However, in this position paper, we discuss how PTLMs like BERT fail to connect simple semantically similar concepts and words, despite given all the necessary context. In Table 1, we purposefully probe BERT to show what kind of high-level semantics are integrated within its framework, by using the predictions of BERT's Masked Language Modelling (MLM) head using HuggingFace [Wolf *et al.*, 2020] (https://huggingface.co/bert-large-uncased). In the first example, we purposefully vary the structure of a sentence that compares *gold* and *silver* and change the MASK to trigger different behaviors. The first example is a simple one, we use two conjunctions: *and, or*, to try and find out what are all other aspects of comparisons that BERT would predict. In these situations, we realize that BERT just predicts the antonym of the word before the conjunction. For this example, the answer we were looking for is: *large, massive, huge and/or weighty*. In the next step, we decrease the level of abstraction by adding more context, in our example *What's heavier, and thus has a higher [MASK], gold or silver?*, we try to orient BERT towards predicting the *mass*, as it is the characteristic that predicts what element is heavier. However, the accurate context was still not enough for predicting the correct information. Additionally, we vary the sentence structure and the [MASK] position to determine whether this could have an effect on the prediction. Except for when the [MASK] is at the end of the sentence, where the prediction is mainly punctuation, having more context did not help on improving the quality of the prediction, but the predicted vocabulary stayed within the same range. We also experimented with different scenarios including enumerations like in "gold, [MASK] and iron" and "[MASK] such as gold, silver and copper", and in both examples, BERT had

an excellent prediction. This shows that BERT is good when generalizing when it comes to hypernyms and hyponyms, but fails at predicting words in other relations.

We examine multiple examples, varying the topics and vocabulary, and show typical effects from a much larger set of what we have probed. In the second example, we change the topic and try to check which country is bigger and thus has a larger *area* (we would like to note that Wikidata contains the area aspect for these countries). Similarly to the first example, we probe BERT by experiment with sentence structures and MASK positions. The outcomes and our conclusion are identical to the ones from our first example. In the last example set, we keep the same structure for the sentence, and only change the city, the predictions seems to be a bit random and we few similarities between all predictions. These examples are available on Wikidata and Wikipedia, and the information that we are seeking is easily available. In fact, we argue that given the importance of PTLMs and their expensiveness, we should be focusing on including this information to help huge models like BERT capture it in a more systematic way. There has been a rising interest in probing these huge models, and we argue that, much more could be done if we focus on probing PTLMs semantically for various semantic relations, in addition to syntax, morphology and commonsense.

## 3 Conclusion and Future Work

Our goal in this position paper is to probe PTLMs such as BERT to show that they are unable to capture semantic similarity between different words that refer to the same concepts. PTLMs have been shown to improve the quality of many tasks and are not easy to train, which is why there has been a lot of work lately to probe their content for linguistics, syntax and commonsense, and thus make them more useful. Our probing experiments suggest that there could be room for improvement if we enable PTLMs to capture more semantically based information by fine-tuning towards more semantics-oriented objectives. All our examples are extracted from Wikidata to show that, resources such as Wikidata are rich and could be used as a resource for fine-tuning BERT towards more high-level semantics.

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
