# OpenReview forum: "How can BERT Understand High-level Semantics?"
_ijcai.org/IJCAI/2021/Workshop/NSNLI — NSNLI Oral_

### Official Review · Reviewer_w5yh · 2021-05-25
**Very relevant to workshop theme, but lacks detail**

**Rating:** 7
**Confidence:** 4

**Review:**

This position paper presents some examples that seek to sho that pre-trained large language models like BERT etc. are unable to capture "high-level" semantics; and that such failure can be corroborated via a masking scheme to test the outputs of the models in a prediction-style task. The paper is certainly very relevant to the stated topic and goals of the workshop.

Some brief comments:

There exists a lot of prior work on probing LLMs, some of which is recounted in the introduction to this paper - it would be nice to get a succinct grounding of where the current paper differs and what it offers.

The meat of the paper is in the various examples in Table 1 - however, while reading the paper, it took me a while to understand exactly what the "gold" or human ground truth for each of these examples was supposed to be. Some more detail on this would be good to have.

Finally, given that the main contribution of the paper is via Table 1, it would be nice if the authors could share any resources and/or code and techniques that they used in order to come up with such examples, such that their idea can be scaled up and applied to various datasets, etc.

In summary, this paper would make for an interesting contribution to the workshop, and is sure to generate discussion.

---

### Official Review · Reviewer_417N · 2021-05-25
**Topically relevant. Some open questions.**

**Rating:** 5
**Confidence:** 4

**Review:**

The paper argues that pre-trained LMs are unable to capture high-level semantics in language. The paper presents a list of probes that have been tried to demonstrate this claim.

My main concerns are two-fold:

1. It is unclear which BERT model has been used. In fact, I tried the motivating example from Fig 1 on BERT-Large and found that it can predict the expected token "rich". Here's a link: https://huggingface.co/bert-large-uncased?text=We+need+the+net+worth+to+know+how+%5BMASK%5D+he+is.
That said, I also think that the probes are a bit ungrammatical -- which can have a big impact on the performance of the models. Modifying the prompt for the gold-silver example yields the expected result:
https://huggingface.co/bert-large-uncased?text=Due+to+its+higher+%5BMASK%5D%2C+gold+is+heavier+than+silver.
This further raises the point of how these prompts were designed. It'd have been nice if the paper presented a systematic method for coming up with these prompts and examples.

2. Related to the above point, I also think at least one more PTLM needs to be tried before concluding that PTLMs are bad at capturing high-level semantics. (I agree the title specifically mentions BERT. So this point is more and about the goal as mentioned in the main paper.)

---

### Decision · Program_Chairs · 2021-05-27

**Decision:**

Accept (Oral)

**Comment:**

The topic is clearly relevant for the workshop.
Please take the reviewers' comments into account when preparing the camera-ready version.